# DiffTune-PI Based Vector Control of Doubly Fed Induction Generator for Grid-connected Operation

1st Zhiwei Li
School of Marine Electrical Engineering
Dalian Maritime University
Dalian,China
lzw1120220143@dlmu.edu.cn

2nd Gaihui Wang
Changjiang Survey Planning Design
and Research Co.,Ltd
Wuhan,China
975274257@qq.com

3rd Dan Wang*
School of Marine Electrical Engineering
Dalian Maritime University
Dalian,China
dwangdl@gmail.com

4th Liyu Lu
School of Marine Electrical Engineering
Dalian Maritime University
Dalian,China
lly1131@126.com

5th Zhouhua Peng
School of Marine Electrical Engineering
Dalian Maritime University
Dalian,China
zhpeng@dlmu.edu.cn

6th Haoliang Wang
School of Marine Electrical Engineering
Dalian Maritime University
Dalian,China
haoliang.wang12@dlmu.edu.cn

*Abstract*—Due to the unstable feature of wind energy, the fixed parameter control cannot meet the demand of doubly fed induction generator (DFIG) system for wind power generation well. In this paper, for the control of DFIG under grid-connected operation conditions, a vector control method with stator voltage orientation based on DiffTune-PI algorithm is proposed. The control parameters of the PI controller are optimized online by the DiffTune algorithm and the disturbance term is compensated by feed-forward compensation. Simulation results show that the proposed method can effectively suppress the power and voltage fluctuations of the DFIG system under grid-connected operation conditions with excellent dynamic and static performances when generator starts, the grid voltage drops and the input wind speed drops.

*Index Terms*—Doubly Fed Induction Generator (DFIG), DiffTune, Vector Control, Stator Voltage Orientation, Parameter Adaptive Control

## I. INTRODUCTION

In recent years, the utilization of new energy has received more and more attention around the world, and the vigorous development of new energy generation technology is of great significance in solving the increasingly serious problems of energy scarcity and environmental pollution [1-2]. Doubly-fed induction generator (DFIG) is characterized by low converter capacity and the ability to maintain constant frequency at different speeds, such that it is widely used in wind power generation and other fields. However,wind energy is marked by randomness and instability [3], can cause power fluctuations during grid integration, disrupting the balance between grid power and load, and posing blackout risks. Such that minimizing the power oscillation is crucial [4]. Moreover, the proliferation of power electronics in DFIG systems enhances sensitivity to grid voltage and frequency fluctuations [5-6], necessitating advanced wind turbine control strategies.

Common DFIG control strategies include sliding mode control [7], model predictive control [8], vector control [9], etc. Among them, many scholars have studied the vector control, a classical control method. In [10], a dual loop vector control strategy is proposed, which uses a current loop and a flux loop to control the active power and reactive power, such that an almost constant stator active power and electromagnetic torque are obtained . In [11], compensation is added to the reference current and a vector control for stator flux orientation is proposed. The oscillations of torque, active power and reactive power can be effectively reduced. In [12], a combined vector and direct power control is proposed, which has the advantages of both vector control (VC) and direct power control methods: fast dynamic response, low computation, power ripple and distortion. A direct stator current vector control method without a phase-locked loop (PLL) is proposed in [13], which calculates the reference values of stator currents using instantaneous power theory, minimizing dependence on generator parameters and ensuring smooth generation of the stator voltage.

However, most of the controllers in the above literatures used fixed parameters, and better control performances may be obtained if the controller parameters can be varied with wind energy conditions. Therefore, in this paper, a stator voltage orientatied vector control method based on the DiffTune-PI algorithm is designed to optimize the control parameters of the PI controller online through the DiffTune algorithm. And the perturbation term is compensated using feedforward control. The robustness of the DFIG system to power fluctuations and voltage drops under grid-connected conditions is enhanced. Compared with the traditional control methods, it has superior

This work was supported by the National Natural Science Foundation of China under Grants 52071044, 51979020. Key Basic Research of Dalian under Grant 2023JJ11CG008 and in part by the Liaoning Revitalization Talents Program under Grant XLYC2007188, in part by the Dalian High-level Talents Innovation Support Program under Grant 2022RQ010, in part by the Fundamental Research Funds for the Central Universities 3132023508. Bolian Research Funds of Dalian Maritime University/Fundamental Research Funds for the Central Universities under Grant 3132023616, the Open Project of State Key Laboratory of Maritime Technology and Safety under Grant SKLMTA-DMU2024Y3.

dynamic response and transient performance.

## II. MATHEMATICAL MODEL OF DFIG

The block diagram of the DFIG system for grid connected operation is shown in Fig. 1. The wind turbine is connected to the gearbox to drive the rotor of the DFIG, and the achieved power is transferred from the DFIG stator winding to the transformer and then to the grid; the DFIG rotor winding is connected to the stator winding through two back-to-back power converters, including the stator-side rectifier and the rotor-side inverter. Stator-side rectifier is mainly used to maintain the capacitor voltage stable, and the rotor-side inverter provides the excitation current for the DFIG to control its grid-connected power generation operation.

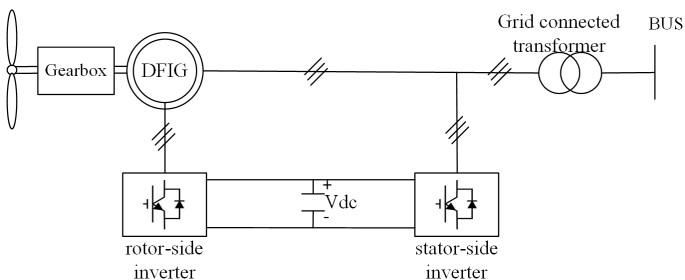

Fig. 1. Block diagram of DFIG system for grid connected operation.

According to motor convention, the mathematical model of DFIG system in a synchronous rotating coordinate system is

$$\begin{cases} \Psi_{sd} = L_s i_{sd} + L_m i_{rd} \\ \Psi_{sq} = L_s i_{sq} + L_m i_{rq} \\ \Psi_{rd} = L_m i_{sd} + L_r i_{rd} \\ \Psi_{rq} = L_m i_{sq} + L_r i_{rq} \end{cases} \tag{1}$$

$$\begin{cases} u_{sd} = d\Psi_{sd}/dt + R_s i_{sd} - \omega_s \Psi_{sq} \\ u_{sq} = d\Psi_{sq}/dt + R_s i_{sq} + \omega_s \Psi_{sd} \\ u_{rd} = d\Psi_{rd}/dt + R_r i_{rd} - \omega_{sl} \Psi_{rq} \\ u_{rq} = d\Psi_{rq}/dt + R_r i_{rq} + \omega_{sl} \Psi_{rd} \end{cases} \tag{2}$$

where $i$, $u$, $\Psi$ represents current, voltage, and magnetic flux, respectively. The subscripts $d$ and $q$ represent the d and q axes of the synchronous coordinate system, and the subscripts s and r represent the parameters of the stator and rotor. $R_s$ and $R_r$ are the stator resistance and rotor resistance, respectively; $L_s$, $L_r$, and $Lm$ represent stator inductance, rotor inductance, and mutual inductance between stator and rotor, respectively; $\omega_s$ and $\omega_{sl}$ are the stator angular velocity and slip angular velocity of the DFIG, respectively. $\omega_r$ is the electrical frequency corresponding to the rotor speed of the DFIG. In steady state operation, the relationship among $\omega_s$, $\omega_r$ and $\omega_{sl}$ is

$$\omega_s = \omega_r + \omega_{sl} \tag{3}$$

## III. VECTOR CONTROL AND DIFFTUNE-PI

### A. Stator Voltage Vector Orientatied Vector Control

The stator voltage orientation is to align the q-axis of the synchronous reference coordinate system with the stator voltage vector, and is rotated clockwise 90 degrees in the direction of the d-axis. The dq coordinate system is rotated at the same speed as the voltage vector, as shown in Fig. 2.

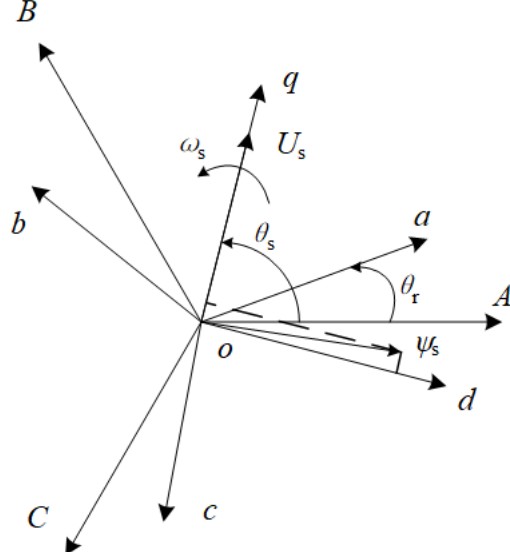

Fig. 2. Stator voltage oriented synchronous rotating coordinate system.

The stator voltage vector in the $dq$ synchronous rotating coordinate system can be expressed as

$$U_{sdq} = u_s e^{-j\widetilde{\theta}_s} \tag{4}$$

where $U_{sdq}$ is the stator voltage vector in the dq synchronous rotation coordinate system, $u_s$ is the amplitude of the stator voltage vector, and $\widetilde{\theta}_s$ is the difference between the synchronized rotation angle $\theta_s$ and the estimated value.

When the synchronous rotating coordinate system is accurately oriented, $\widetilde{\theta}_s = 0$, (4) can be rewritten as

$$U_{sdq} = u_s \tag{5}$$

This means that in synchronous rotating coordinate system with stator voltage vector orientation, the stator voltage vector is located on the q-axis of the coordinate system, such that

$$\begin{cases} u_{sq} = u_s \\ u_{sd} = 0 \end{cases} \tag{6}$$

From (1), it can be concluded that

$$\begin{cases} i_{sq} = (\Psi_{sq}/L_s) - i_{rq}L_m/L_s \\ i_{sd} = (\Psi_{sd}/L_s) - i_{rd}L_m/L_s \end{cases} \tag{7}$$

By substituting (6) into the rotor flux equation of a DFIG, it can be obtained that

$$\begin{cases} \Psi_{rq} = (L_r - (L_m^2/L_s))i_{rq} + \Psi_{sq}L_m/L_s \\ \Psi_{rd} = (L_r - (L_m^2/L_s))i_{rd} + \Psi_{sd}L_m/L_s \end{cases} \quad (8)$$

The stator resistance of DFIG, especially high-power DFIG at the megawatt level, can usually be ignored compared to their inductance. When the stator resistance is ignored, substitute (6 - 8) into (2), it can be obtained that

$$\begin{cases} u_{rq} = R_r i_{rq} + \sigma L_r(di_{rq}/dt) + (L_m/L_s)u_s \\ \quad -(L_m/L_s)\omega_r\Psi_s sd + \omega_{sl}\sigma L_r i_{rd} \\ u_{rd} = R_r i_{rd} + \sigma L_r(di_{rd}/dt) - (L_m/L_s)\omega_r\Psi_{sq} \\ \quad -\omega_{sl}\sigma L_r i_{rq} \end{cases} \quad (9)$$

In the formula, $\sigma = (L_s L_r - L_m^2)/(L_s L_r)$ is the leakage magnetic coefficient. In the case of stator voltage vector orientation, the expression for active and reactive power on the stator side of a DFIG is

$$\begin{cases} P_s = 1.5 u_s i_{sq} \\ Q_s = 1.5 u_s i_{sd} \end{cases} \quad (10)$$

Substituting (7) into (10) yields

$$\begin{cases} P_s = 1.5 u_s(\Psi_{sq} - L_m i_{rq})/L_s \\ Q_s = 1.5 u_s(\Psi_{sd} - L_m i_{rd})/L_s \end{cases} \quad (11)$$

Equation (10) indicates that under the condition of constant stator voltage and stator flux, such that during steady-state operation, the active power $P_s$ on the stator side of the DFIG is mainly determined by the q-axis component $i_{rq}$ of the rotor current, while the reactive power $Q_s$ is mainly determined by the d-axis component $i_{rd}$ of the rotor current.

If PI control is applied to the rotor current of the DFIG and the transfer function of the PI controller is made to be $k_P + k_I/s$, such that, the PI controller is used to control the dynamic term of the rotor current in (11), while the feedforward compensation algorithm is applied to the disturbance term, then the rotor voltage control equation is as follows

$$\begin{cases} u_{rq} = (k_P + (k_I/s))(i_{rq}^* - i_{rq}) + u_{rqc} \\ u_{rd} = (k_P + (k_I/s))(i_{rd}^* - i_{rd}) - u_{rdc} \end{cases} \quad (12)$$

$$\begin{cases} u_{rqc} = (L_m/L_s)u_s - (L_m/L_s)((R_s/L_s)\Psi_{sq} + \omega_r\Psi_{sd}) \\ \quad +\omega_{sl}(L_r - L_m^2/L_s)i_{rd} \\ u_{rdc} = (L_m/L_s)((R_s/L_s)\Psi_{sd} + \omega_r\Psi_{sq}) \\ \quad +\omega_{sl}(L_r - L_m^2/L_s)i_{rq} \end{cases}$$
$$(13)$$

### B. DiffTune-PI Control Algorithm

The operation of DFIG is a dynamic process, and the PI control with fixed parameters cannot fully adapt to the changing operation status of DFIG. Such that a gradient-based parameter adaptive algorithm proposed in [14]. The algorithm is called "DiffTune algorithm", which can be used to optimize the controller parameters online for better control performances.

In this section, only the q-axis parameter control of DFIG is explained, since the d-axis parameter control method is similar.

Discretizing the DFIG system, the equations for $i_{rq}$ can be obtained as

$$i_{rq_{n+1}} = i_{rq_n} + (T_s/L_r)(u_{rq_n} - R_r i_{rq_n}) \quad (14)$$

where $T_s$ is the sampling interval and $X_n$ represents the value of $X$ at the moment n. According to (11), it can be obtained that

$$u_{rq_n} = (k_P + (k_I/s))(i_{rq_n}^* - i_{rq_n}) + u_{rqc_n} \quad (15)$$

Thus, the equations for $i_{rq}$ and $u_{rq}$ can be obtained as

$$i_{rq_{n+1}} = f(i_{rq_n}, u_{rq_n}) \quad (16)$$

$$u_{rq_n} = h(i_{rq_n}^*, i_{rq_n}, u_{rq_n}, k) \quad (17)$$

where $k \in R$ denotes the controller parameters, such that, $k_P$, $k_I$. The initial state $i_{rq0}$ is known, the state $i_{rq}$ is directly measurable, and the state (16) and the controller (17) are both differentiable.

The minimization evaluation criterion used for adjusting $k$ is (18), abbreviated as $L(k)$. It is a differentiable function of the reference state $i_{rq}^*$, the actual state $i_{rq}$, and the control quantity $u_{rq}$ over the time interval $T$, where $\lambda > 0$ is the penalty factor.

$$L(i_{rq0:T}, i_{rq0:T}^*, u_{rq0:T-1}; k) = \sum_{n=0}^{T} \|i_{rqn} - i_{rqn}^*\|^2 + \sum_{n=0}^{T-1} \lambda \|u_{rqn}\|^2$$
$$(18)$$

Gradient based computation is used to minimize $L(k)$. The controller parameters are regulated using gradient descent to optimize the parameters $k_P$, $k_I$ and improve the system's performance.

The gradient $\nabla_k L$ is used to stepwise optimize the parameter $k$. The parameter $k$ is generally in the feasible domain $K$, such that $k$ can be optimized by using projected gradient descent [15].

$$k \leftarrow P_K(k - \beta \nabla_k L) \quad (19)$$

In (19), $P_K$ is the projection operator used to project the operands into the feasible domain $K$ and $\beta$ is the rate of change. The stability of the system is ensured by using the feasible domain $K$, which can be determined either by practical engineering experience or by Liapunov analysis.

Since the DFIG system in this paper has no special constraints, the projected gradient descent method can be simplified to the ordinary gradient descent method.

$$k \leftarrow (k - \beta \nabla_k L) \quad (20)$$

The sensitivity propagation method was used to compute the gradient $\nabla_k L$. The gradient $\nabla_k L$ can be expanded and written as

$$\nabla_k L = \sum_{n=1}^{T} (\partial L/\partial i_{rqn})(\partial i_{rqn}/\partial k)$$
$$+ \sum_{n=0}^{T-1} (\partial L/\partial u_{rqn})(\partial u_{rqn}/\partial k) \quad (21)$$

Since $L$ has been identified in (18), it can be calculated that $\partial L/\partial i_{rqn}$ and $\partial L/\partial u_{rqn}$ are

$$\begin{cases} \partial L/\partial i_{rqn} = 2(i_{rqn} - i_{rqn}^*) \\ \partial L/\partial u_{rqn} = 2u_{rqn} \end{cases} \quad (22)$$

Taking partial derivatives with respect to $k$ on both sides of (16) and (17) yields

$$\partial i_{rqn+1}/\partial k = (\nabla_{i_{rqn}} f + \nabla_{u_{rqn}} f \nabla_{i_{rqn}} h)(\partial i_{rqn}/\partial k) + \nabla_{u_{rqn}} f \nabla_k h \quad (23)$$

$$\partial u_{rqn}/\partial k = \nabla_{i_{rqn}} h(\partial i_{rqn}/\partial k) + \nabla_k h \quad (24)$$

This is the iterative computational equation for $\partial i_{rqn}/\partial k$ and $\partial u_{rqn}/\partial k$, where $\partial i_{rq0}/\partial k = 0$. $\partial i_{rqn}/\partial k$ and $\partial u_{rqn}/\partial k$ can be reffered to as sensitivity states. Eq. 23 can be considered as a time-varying linear system with sensitivity state $\partial i_{rqn}/\partial k$. By sampling the DFIG system, the state $i_{rqn}$ and control $u_{rqn}$ are obtained for the calculation of the system coefficients $\nabla_{irq}f, \nabla_{urq}f, \nabla_{irq}h, \nabla_{urq}h$. And $\nabla_k L$ can be considered as a weighted sum of weights $(\partial L/\partial i_{rqn})_{n=0:T}$ and $(\partial L/\partial u_{rqn})_{n=0:T-1}$ for the $(\partial i_{rqn}/\partial k)_{n=0:T}$ and $(\partial u_{rqn}/\partial k)_{n=0:T-1}$ .

Because $\partial i_{rqn+1}/\partial k$ can be updated online through sensitivity propagation whenever the system data $i_{rqn}$ and $u_{rqn}$ are updated, and the sampling time $T$ can be changed online as needed, the sensitivity propagation method can be adapted online, which is highly flexible for changing systems.

Therefore, the PI controller for optimizing the parameters using the DiffTune algorithm can be designed as Fig. 3.

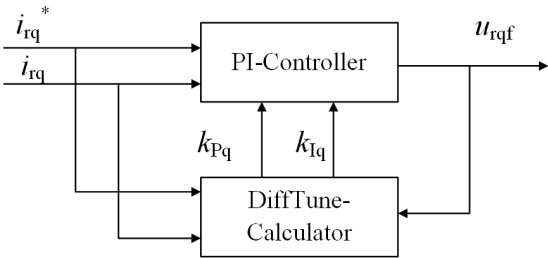

Fig. 3. DiffTune-PI Controller.

## IV. SIMULATION RESULTS

Matlab simulations of a 1.5 $MW$ DFIG generation system are presented to verify the effectiveness and feasibility of the proposed method. The block diagram of the simulation system is shown in Fig. 4.

The power controller and rotor current controller are applied in the inner and outer loop controllers of the grid-connected

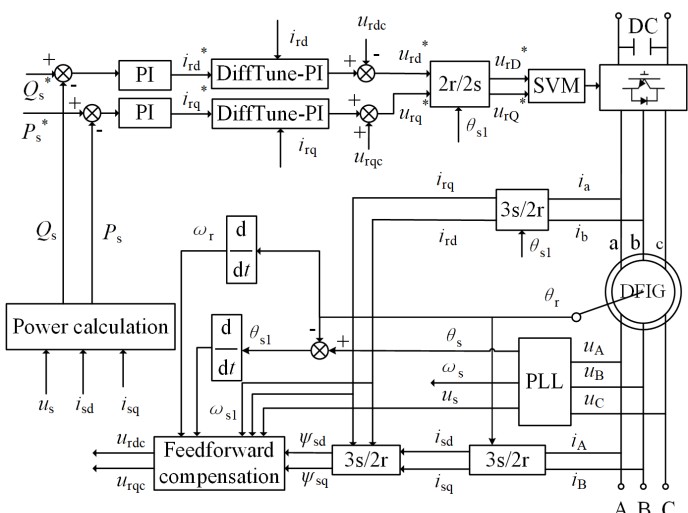

Fig. 4. Stator voltage orientatied control structure of DFIG.

DFIG power generation system. The stator voltage, stator current, rotor current and rotor position signals are sampled , respectively, which are inputted into the inner and outer loop controllers to generate SVM signals. Then SVM signals are sent to the rotor-side inverter to control the grid-connected operation of the doubly-fed generator. The simulation parameters of this system are set as follows: AC-grid-connected voltage is given as 690V, the DC side voltage is given as 1200V, the angular frequency $\omega^*$ is given as $100\pi rad/s$; and the stator frequency is 50Hz. The prime mover is a wind turbine with a rated power of 1.5 MW. Other parameters of DFIG are shown in Table 1.

TABLE I
PARAMETERS OF DFIG

| Parameter | Symbol | Simulation |
|---|---|---|
| power rating | $P_N$ | $1.5MW$ |
| number of machine poles | $n_p$ | 2 |
| stator resistance | $R_s$ | $5.5m\Omega$ |
| rotor resistance | $R_r$ | $6.21m\Omega$ |
| stator inductance | $L_s$ | $0.156mH$ |
| rotor inductance | $L_r$ | $0.226mH$ |
| mutual inductance | $L_m$ | $11.01mH$ |

In order to verify the effectiveness of the proposed algorithm during grid-connected operation, the DiffTune-PI control method proposed in this paper and the fixed-parameter PI control method are compared under three operating conditions: generator starting process, a drop in grid voltage and drop in input wind speed.

The fixed-parameter PI controller's parameters, $k_{Pd}$, $k_{Id}$, $k_{Pq}$, $k_{Iq}$, are set to 0.135, 1.700, 0.090, 1.800 , respectively.

### A. Response to generator starting process

Fig. 5 illustrates the optimization of the DiffTune-PI controller parameters over time. It can be seen that the parameters of the PI controller, $k_{Pd}$, $k_{Id}$, $k_{Pq}$, $k_{Iq}$, keep changing during

the generator starting process. After 0.4s, they stabilize at 0.159, 1.620, 0.094, 1.035, respectively. Fig. 6 shows the variation of $U_{DC}$ from generator startup to stabilization in 0 to 0.6 seconds. It can be seen that when the fixed-parameter PI control is used, the maximum inrush voltage is as high as 1957 V and the $U_{DC}$ is almost always substantially higher than the given value of 1200 V until 0.15 s. In contrast, when the DiffTune-PI algorithm is used, the maximum inrush voltage is only 1663 V and the $U_{DC}$ is relatively close to the given voltage throughout the transition to stable operation. This shows that DiffTune, by optimizing the control parameters of the PI controller online, can make the system more adaptable to the state changes of the startup process and achieve better control results.

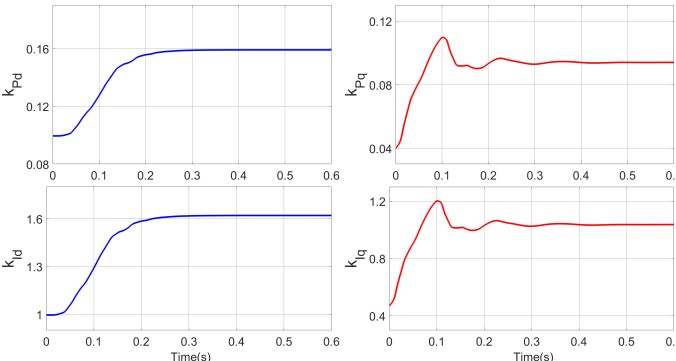

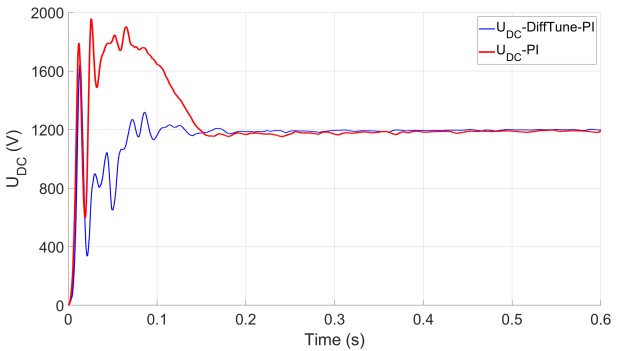

Fig. 5. Parameters of DiffTune-PI Controller.

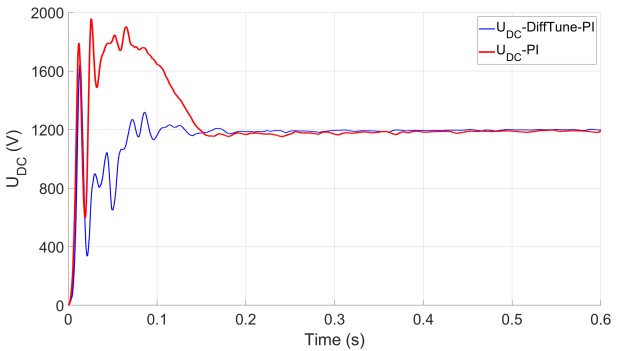

Fig. 6. $U_{DC}$ Control Comparison.

### B. Response to grid voltage drop

It is assumed that the grid voltage decreases to 80% of the original one, such that, 552 V, at 0.6 s. Fig. 7 shows the parameter variation curve of the Difftune-PI controller, It can be seen that the parameters of the PI controller, $k_{Pd}$, $k_{Id}$, $k_{Pq}$, $k_{Iq}$, keep changing after the grid voltage decreases. After 0.9s, they stabilize at 0.179, 1.828, 0.090, and 0.990, respectively. Fig. 8 shows the variation curve of the DC-side voltage $U_{DC}$. From Fig. 9, it can be seen that using fixed-parameter PI control, the $U_{DC}$ is as low as 1104 V. While using Difftune-PI control, the $U_{DC}$ is only as low as 1130 V,

which is 2.5% higher than that of fixed-parameter PI control. It shows that Difftune-PI has more robustness in the face of grid voltage fluctuation.

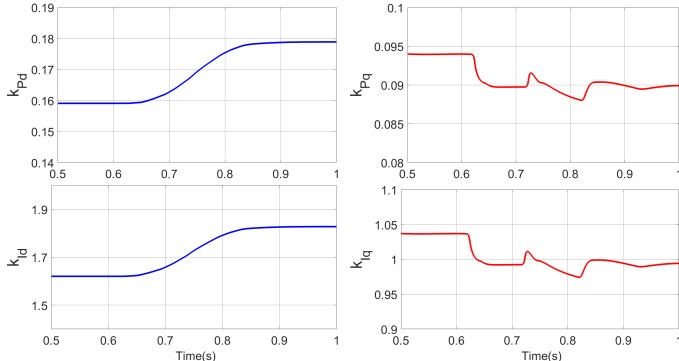

Fig. 7. Parameters of DiffTune-PI Controller.

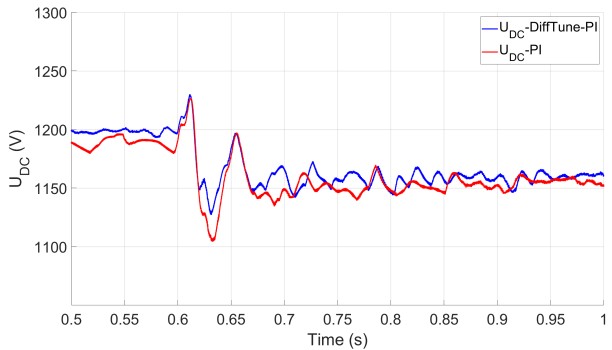

Fig. 8. $U_{DC}$ Control Comparison.

### C. Response to input wind speed drop

It is assumed that the wind speed input to the wind turbine is reduced from the rated wind speed of 11.5m/s to 10.05m/s at 0.6s, such that the output power is reduced from 1.5MW to 1MW. Fig. 9 shows the parameter variation curves of the Difftune-PI controller, It can be seen that the parameters of the PI controller, $k_{Pd}$, $k_{Id}$, $k_{Pq}$, $k_{Iq}$, keep changing after the wind speed decreases. After 0.9s, they stabilize at 0.186, 1.906, 0.093, and 1.031, respectively. Fig. 10 shows the variation curves of the active power output of the DFIG. When the the input wind speed drops, Difftune-PI control and fixed-parameter PI control have similar control effects on $U_{DC}$. However,as can be seen in Fig. 10, using the fixed-parameter PI control, the minimum output power is 0.64MW, and the output power stabilizes after 0.73s. While using the DiffTune-PI control, the lowest output power is 0.71MW, which is 11% higher than the fixed parameter PI control. The output power stabilizes after 0.69s, and the output power fluctuation is smaller. It shows that using DiffTune-PI control can make the DFIG system transition to the new state of outputting 1MW power more quickly and smoothly.

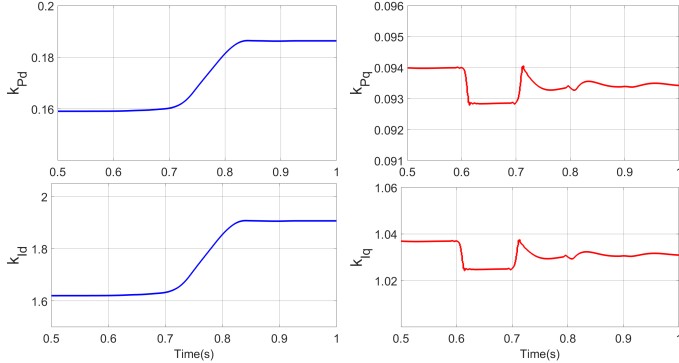

Fig. 9. Parameters of DiffTune-PI Controller.

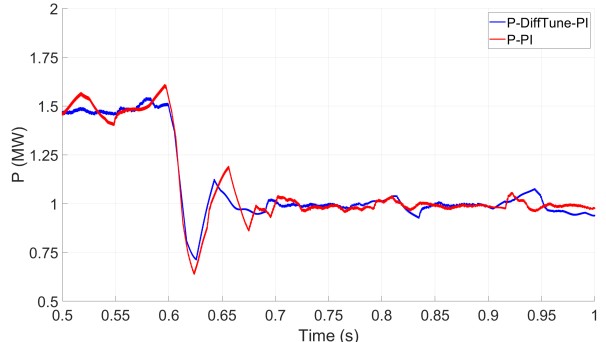

Fig. 10. Active Power Comparison.

## V. CONCLUSION

In this paper, a vector control method based on stator voltage orientation, DiffTune-PI control and feed-forward control is proposed for the control of DFIG operating under grid-connected conditions. The parameters of the PI controller are optimized online by the DiffTune algorithm to control the dynamic term of the rotor current and the feed-forward compensated control of the rotor current disturbance term, which reduces the power shock and voltage shock of the system under three operating conditions: generator starting process, a drop in grid voltage and drop in input wind speed. Simulation results show that the proposed method has stronger robustness and excellent dynamic response performance compared with the fixed-parameter PI control.

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
