# OpenReview forum: "DiffTune-PI Based Vector Control of Doubly Fed Induction Generator for Grid-connected Operation"
_IEEE.org/ICIST/2024/Conference — IEEE ICIST 2024 Conference Submission_

### Official Review · Reviewer_ZJfR · 2024-08-21
**accept**

**Rating:** 7
**Confidence:** 3

**Review:**

This study discussed a vector control method with stator voltage orientation based on DiffTune-PI algorithm for the control of DFIG under grid-connected operation conditions. The theory is correct and can be accepted after responding the following comments.
(1)The introduction merely states the current work, which is not complete, and should be supplemented.
(2) What is the contribution of the paper? It should be highlighted both in the introduction and in the content.
(3)There are many typos and grammar errors. The authors should have a native English speaker or software packages to perform the editing check.
(4)Please explain in detail the role of u and i in the formula.

---

### Official Review · Reviewer_A9oG · 2024-08-22
**This article is very interesting and a good one**

**Rating:** 8
**Confidence:** 3

**Review:**

This study introduced a vector control method with stator voltage orientation based on DiffTune-PI algorithm for the control of DFIG under grid-connected operation conditions. The theory is correct and can be accepted after responding the following comments.
(1) In the introduction, it is not enough to state the current work. It should be expended and reconstructed.
(2) In the end of Section 1, the organization of this study is suggested to be summarized.
(3) Could you elaborate on the roles that $u$ and  $i$ play in formula (2), and clarify whether they are explicitly represented in formulas (1) and (2), or if their presence differs?
(4) In the simulation section, more analysis can be added to better explain the main results of this paper, that's not enough.
(5) There are many typos and grammar errors. The authors should have a native English speaker or software packages to perform the editing check.
(6) The future work is missing in the Conclusion.

---

### Official Review · Reviewer_A6Bw · 2024-08-24
**In this paper, for the control of DFIG under grid-connected operation conditions, a vector control method with stator voltage orientation based on DiffTune-PI algorithm is proposed.  The topic of this paper is interesting. Below is a list of comments that should be taken into account further when revising the paper.**

**Rating:** 7
**Confidence:** 3

**Review:**

1. The contribution of this article should be compared with previous literature, and the basic technical difficulties of this article should be listed? And what methods should be used to solve this problem, emphasizing novelty and technological contribution.
2. In the simulation results section, the horizontal axis time of the simulation diagram should be longer, which ensures that readers can have a more intuitive understanding of the advantages of this method.
3. In the conclusion section, the author should elaborate on the proposed vector control method based on stator voltage direction, DiffTune-PI control, and feedforward control for controlling DFIG operating under grid connected conditions. Meanwhile, please provide a detailed explanation of your future plans.

---

### Comment · Reviewer_A9oG · 2024-08-21
**This article is very interesting and a good one**

This study introduced a vector control method with stator voltage orientation based on DiffTune-PI algorithm for the control of DFIG under grid-connected operation conditions. The theory is correct and can be accepted after responding the following comments.
(1)	In the introduction, it is not enough to state the current work. It should be expended and reconstructed.
(2)	In the end of Section 1, the organization of this study is suggested to be summarized.
(3)	Could you elaborate on the roles that '$u$' and '$i$' play in formula (2), and clarify whether they are explicitly represented in formulas
        (1) and (2), or if their presence differs?
(4)	In the simulation section, more analysis can be added to better explain the main results of this paper, that's not enough.
(5)	There are many typos and grammar errors. The authors should have a native English speaker or software packages to perform the
        editing check.
(6)	The future work is missing in the Conclusion.

---

### Decision · Program_Chairs · 2024-09-06

Accept (Oral)